# The first national survey of antimicrobial use among dentists in Japan from 2015 to 2017 based on the national database of health insurance claims and specific health checkups of Japan

Akane Ono[1], Masahiro Ishikane[1,2], Yoshiki Kusama[1]*, Chika Tanaka[1,2], Sachiko Ono[3], Shinya Tsuzuki[1], Yuichi Muraki[4], Daisuke Yamasaki[5], Masaki Tanabe[5], Norio Ohmagari[1,2]

1 AMR Clinical Reference Center, National Center for Global Health and Medicine, Tokyo, Japan, 2 Disease Control and Prevention Center, National Center for Global Health and Medicine, Tokyo, Japan, 3 Department of Eat-loss Medicine, The University of Tokyo, Tokyo, Japan, 4 Department of Clinical Pharmacoepidemiology, Kyoto Pharmaceutical University, Kyoto, Japan, 5 Department of Infection Control and Prevention, Mie University Hospital, Mie, Japan

☯ These authors contributed equally to this work.
* stone.bagle@gmail.com

**Data Availability Statement:** Data cannot be shared publicly because of including personal

## Abstract

### Purpose

To counter the global health threat of antimicrobial resistance, effective antimicrobial stewardship programs are needed to improve antimicrobial use (AMU) among dentists in addition to physicians. This study aimed to investigate the nationwide epidemiology of AMU among Japanese dentists to facilitate the development of dentist-centered programs.

### Methods

We conducted a retrospective population-based study using the National Database of Health Insurance Claims and Specific Health Checkups of Japan to analyze the AMU among Japanese dentists between 2015 and 2017. AMU was quantified as the defined daily doses per 1,000 inhabitants per day (DID). The trends in dentist-prescribed AMU were examined according to antimicrobial category and administration route. We also compared outpatient oral AMU between dentists and physicians as well as between on-site and off-site dispensing.

### Results

The DID values of dentist-prescribed AMU were 1.23 in 2015, 1.22 in 2016, and 1.21 in 2017. During this study period, outpatient oral antimicrobials comprised the majority (approximately 99%) of dentist-prescribed AMU, and cephalosporins were the most frequently prescribed antimicrobials (>60% of all antimicrobials). The DID values of outpatient oral AMU were 1.21 for dentists and 12.11 for physicians. The DID value for on-site

information. Data are available from the Ministry of Health, Labour, and Welfare of Japan for researchers who meet the criteria for access to confidential data. Contact information is below: Phone +81-50-5546-9167 email teikyo_rezept@kits.nttdata.co.jp.

**Funding:** This study was supported by a research grant from the Ministry of Health, Labour and Welfare of Japan (Grant Number: 20HA2003).

**Competing interests:** Yuichi Muraki received an honorarium for lecturing from Pfizer Japan, Inc. The other authors have no conflicts of interest to declare. This does not alter our adherence to PLOS ONE policies on sharing data and materials.

dispensing was 0.89 in 2017, in which cephalosporins were the predominantly used antimicrobials (DID: 0.60).

## Conclusions

Interventions that target dentists in Japan should focus on on-site dispensing of oral antimicrobials (especially cephalosporins) for outpatients. Further studies are needed to ascertain the underlying factors of oral cephalosporin prescriptions to guide the development of effective antimicrobial stewardship programs.

## Introduction

The World Health Organization has advocated that each country develops a national action plan to counter the rising global health threat of antimicrobial resistance (AMR). Such plans include targeted reductions in antimicrobial use (AMU) within predetermined time frames and the promotion of antimicrobial stewardship programs (ASPs). Several countries have reported that dentists account for 7–11% of all AMU [1–6]. In the US, dentists constitute the second-leading prescribers of antimicrobials (10%) after physicians (80%) [7], with outpatient prescriptions forming the majority (59.1%) of antimicrobial-related expenditures [8]. Although AMU in dental practice is much lower than that of medical practice in the US, Germany, and England [1–3, 5, 6], a rise in dentist-prescribed AMU has been reported in Canada [4]. That study also suggested a need for ASPs that target dentists due to a propensity to unnecessarily prescribe antimicrobials for periapical abscesses and irreversible pulpitis, as well as their sluggish adoption of updated guidelines that stipulate the reduction of perioperative AMU for patients with valvular heart disease and prosthetic joints [4].

Japan's National Action Plan on AMR from 2016 to 2020 showed that its national AMU was comparable to that of other industrialized countries in the EU [9]. However, Japan uses a much larger proportion of broad-spectrum antimicrobials (e.g., cephalosporins, macrolides, and fluoroquinolones) and a smaller proportion of narrow-spectrum agents (e.g., penicillin) than these other countries [9]. These trends can potentially contribute to the spread of AMR in Japan, and there is therefore a need to assess and control the use of broad-spectrum antimicrobials. Two studies with small sample sizes in limited settings have reported on the inappropriate use of antimicrobials by Japanese dentists [10, 11]. One of these studies found that over 80% of dentists prescribed prophylactic cephalosporins for dental procedures [10], and the other showed that dentists often prescribed antimicrobials post-procedurally despite limited supporting evidence for their effectiveness [11]. The design of ASPs should be informed by an understanding of current AMU trends, but no studies have explored the nationwide epidemiology of AMU among dentists in Japan.

Thus, we conducted a retrospective population-based study using data from the National Database of Health Insurance Claims and Specific Health Checkups of Japan (NDB) between 2015 and 2017 to investigate the epidemiology of AMU among Japanese dentists to support the development and implementation of effective ASPs.

## Methods

### Ethics statement

This study was approved by the ethics committee of the National Center for Global Health and Medicine (Approval Number: NCGM-G-002505-00), and was conducted in accordance with

the Declaration of Helsinki. Patient data were anonymized prior to analysis. The ethics committee waived the need for patient consent because the study did not deal with any personally identifiable information.

## Study design and data source

This retrospective population-based study was conducted using NDB data from January 2015 to December 2017. In Japan, the national health insurance system provides universal coverage for all citizens and long-term residents. Insurance claims for almost all national health insurance–covered healthcare goods and services are collected and stored in the NDB after anonymization. From the NDB, we extracted all claims for antimicrobials that were prescribed by dentists during the study period. We only obtained aggregate data that did not include personally identifiable information, such as birth dates. In addition, the NDB only includes electronic claims data (without any paper-based claims). With the progression of data digitization, the proportion of electronic claims from dentists had steadily increased from 40% in 2011 to 96% in 2015 [12]. Therefore, we focused on data from 2015 to 2017 even though NDB data were available from 2011 onward. We also acquired data on antimicrobial prescriptions by physicians in 2016. The numbers of dentists and physicians were obtained from the National Surveys of Physicians, Dentists and Pharmacists published by the Ministry of Health, Labour and Welfare of Japan [13, 14]. The open data on physician- and dentist-prescribed AMU were acquired from the AMR Clinical Reference Center website [15].

Prescriptions were categorized into outpatient and inpatient prescriptions, and outpatient prescriptions were further sub-categorized into on-site and off-site dispensing. On-site dispensing was defined as antimicrobial prescriptions that are filled at the prescribing facility. In contrast, off-site dispensing referred to antimicrobial prescriptions that are filled at a third-party retail pharmacy, which allows the prescribing physician/dentist to prescribe any antimicrobial regardless of whether their facility has it in stock.

## Antimicrobial categories and use

Antimicrobials were categorized based on the World Health Organization's Anatomical Therapeutic Chemicals (ATC) Classification System and administration route (oral vs. parenteral). AMU was standardized as the defined daily dose, and the volume of use was quantified as the defined daily doses per 1,000 inhabitants per day (DID) [16–18]. The number of inhabitants was obtained from annual population estimates published by the Ministry of Internal Affairs and Communications of Japan [19]. We examined AMU according to the following ATC Level 4 categories: penicillins (ATC code: J01C), cephalosporins (J01DB/J01DC/J01DD/J01DE), macrolides (J01FA), fluoroquinolones (J01MA), and others (J01 excluding the aforementioned codes). Although the ATC code J01DI also includes cephalosporins (ceftobiprole medocaril, ceftaroline fosamil, and ceftolozane/β-lactamase inhibitor combination), these were not used in Japan during the study period. Accordingly, J01DI was sorted into the "others" category.

## Outcomes of interest and statistical analysis

We first analyzed the annual trends in dentist-prescribed total AMU stratified by antimicrobial category and administration route between 2015 and 2017. Thereafter, we compared (i) oral AMU in the outpatient setting between dentists and physicians in 2016 (when AMU data from physicians were available) [15] and (ii) oral AMU in the outpatient setting between on-site and off-site dispensing among dentists in 2017. The patterns in oral AMU between dentists and physicians were compared using chi-square tests. Statistical significance was set at $P < 0.05$

**Table 1. Annual total antimicrobial use among dentists from 2015 to 2017 stratified by antimicrobial category.**

| | 2015 | | 2016 | | 2017 | |
|---|---|---|---|---|---|---|
| Penicillins | 0.10 | (8.0) | 0.11 | (8.8) | 0.13 | (10.3) |
| Cephalosporins | 0.81 | (65.6) | 0.80 | (65.2) | 0.77 | (63.7) |
| Macrolides | 0.23 | (19.0) | 0.30 | (18.7) | 0.23 | (18.8) |
| Fluoroquinolones | 0.067 | (5.4) | 0.067 | (5.4) | 0.066 | (5.4) |
| Others | 0.023 | (1.9) | 0.023 | (1.9) | 0.022 | (1.8) |
| Total | 1.23 | | 1.22 | | 1.21 | |

Values are presented as defined daily doses per 1,000 inhabitants per day (% of total).

(two-sided). All statistical analyses were performed using SPSS Statistics Version 25 (IBM Corp., Armonk, NY, US).

## Results

### Annual trends in total AMU among dentists stratified by antimicrobial category and administration route in 2015 to 2017

Table 1 shows the DID values of total AMU among dentists stratified by antimicrobial category. The DID values were 1.23 in 2015, 1.22 in 2016, and 1.21 in 2017. Cephalosporins accounted for the majority of antimicrobials with DID values (proportion of total antimicrobials) of 0.81 (65.6%) in 2015, 0.80 (65.2%) in 2016, and 0.77 (63.7%) in 2017. Over the study period, the DID values of penicillins gradually increased while those of cephalosporins slowly decreased. In contrast, the AMU trends for macrolides and fluoroquinolones were stable over time. The penicillins in Table 1 included antimicrobials with ATC codes J01CA, J01CE, and J01CR. The DID values of J01CA antimicrobials (which include amoxicillin) were 0.10 in 2015, 0.10 in 2016, and 0.12 in 2017. The DID values of J01FF antimicrobials (which include clindamycin) were 0.0017 in 2015, 0.0019 in 2016, and 0.0022 in 2017. Table 2 presents the DID values of total AMU among dentists stratified by administration route. In all three years, outpatient oral antimicrobials comprised the majority of dentist-prescribed AMU (approximately 99%).

### Comparison of outpatient oral AMU between dentists and physicians in 2016

Table 3 shows the outpatient oral AMU for dentists (n = 104,533) and physicians (n = 319,480) in 2016. The DID values were 1.21 for dentists and 12.11 for physicians. In addition, dentists accounted for approximately 9.1% of the national outpatient oral AMU (i.e., outpatient oral AMU for both dentists and physicians). There were significant differences in the proportions of outpatient oral AMU between dentists and physicians ($P < 0.001$).

**Table 2. Annual total antimicrobial use among dentists from 2015 to 2017 stratified by administration route.**

| | 2015 | | 2016 | | 2017 | |
|---|---|---|---|---|---|---|
| Outpatient oral administration | 1.213 | (99.0) | 1.21 | (98.9) | 1.20 | (98.9) |
| Outpatient parenteral administration | <0.010 | (0.1) | <0.010 | (0.1) | <0.010 | (0.1) |
| Inpatient oral administration | <0.010 | (0.5) | <0.010 | (0.5) | <0.010 | (0.5) |
| Inpatient parenteral administration | <0.010 | (0.4) | <0.010 | (0.4) | <0.010 | (0.4) |
| Total | 1.23 | | 1.22 | | 1.21 | |

Values are presented as defined daily doses per 1,000 inhabitants per day (% of total).

**Table 3. Comparison of outpatient oral antimicrobial use between dentists and physicians in 2016.**

|  | Dentists n = 104,533 | | Physicians n = 319,480 | |
| --- | --- | --- | --- | --- |
| Penicillins | 0.11 | (8.7) | 0.98 | (8.1) |
| Cephalosporins | 0.79 | (65.2) | 2.80 | (23.1) |
| Macrolides | 0.23 | (18.9) | 4.47 | (36.9) |
| Fluoroquinolones | 0.067 | (5.5) | 2.69 | (22.2) |
| Others | 0.022 | (1.8) | 1.18 | (9.7) |
| Total | 1.21 | | 12.11 | |

Values are presented as defined daily doses per 1,000 inhabitants per day (% of total).

There were significant differences in the proportions of oral antimicrobial use between dentists and physicians ($P < 0.001$).

## Comparison of outpatient oral AMU between on-site and off-site dispensing among dentists in 2017

Table 4 shows the outpatient oral AMU for on-site and off-site dispensing among dentists in 2017. The DID values were 0.89 for on-site dispensing and 0.31 for off-site dispensing. In addition, on-site dispensing accounted for 74.2% of all outpatient oral AMU among dentists. Cephalosporins were the predominant antimicrobials in both on-site (DID: 0.60; 66.9% of outpatient oral AMU) and off-site (DID: 0.17; 54.1% of outpatient oral AMU) dispensing.

## Discussion

This nationwide study of AMU among Japanese dentists found that the vast majority of their antimicrobial prescriptions were for oral antimicrobials in the outpatient setting (approximately 99%), and that cephalosporins were predominantly used (>60%). These results corroborate those of a previous cross-sectional questionnaire-based study on dentists at community clinics in Japan [20]. The frequent use of cephalosporins in Japan may be due in part to the active promotion of these antimicrobials by pharmaceutical companies. A study conducted in Nepal showed that the top-selling antimicrobials were those that were subjected to intensive promotional activities [21]. Although such promotional activities have since diminished in Japan, the situation was similar to that of Nepal as recently as 10 years ago. These activities may therefore have contributed to the high adoption rate of cephalosporins in Japan's medical facilities. In our previous analysis of AMU in 31 Japanese hospitals, we found that all the study hospitals had adopted the use of oral third-generation cephalosporins [22]. Furthermore, most of these cephalosporins were Japanese-origin drugs that are not commonly used outside of Japan [17]. The prevalent use of cephalosporins by Japanese dentists is in contrast to the utilization patterns in other countries [4, 5, 23–26]. For example, a national survey of German

**Table 4. Comparison of outpatient oral antimicrobial use between on-site and off-site dispensing among dentists in 2017.**

|  | On-site | | Off-site | |
| --- | --- | --- | --- | --- |
| Penicillins | 0.080 | (9.0) | 0.042 | (13.8) |
| Cephalosporins | 0.60 | (66.9) | 0.17 | (54.1) |
| Macrolides | 0.15 | (17.2) | 0.073 | (23.9) |
| Fluoroquinolones | 0.047 | (5.2) | 0.019 | (6.1) |
| Others | 0.014 | (1.6) | 0.006 | (2.1) |
| Total | 0.89 | | 0.31 | |

Values are presented as defined daily doses per 1,000 inhabitants per day (% of total).

dentists showed amoxicillin to be the most commonly prescribed antimicrobial (45.8% of total AMU in 2015) [5]. A Canadian study using a population-based prescribing database reported on the dominant use of amoxicillin (1.26 DID out of 1.59 DID; 79.2% of total AMU in 2013) [4]. Similar findings have also been reported in Kuwait [23] and Scotland [24]. In comparisons of antimicrobial prescription rates, amoxicillin was also found to be the dominant antimicrobial in the US (57% in 2015) [25] and Australia (64.3% in 2016) [26]. Our study also showed that the annual DID value of dentist-prescribed oral AMU was 1.21, which accounted for approximately 9.1% of the national outpatient oral AMU. This proportion was similar to that reported in other countries [4–6].

Although our study found that the AMU of Japanese dentists was lower than that of physicians, the former predominantly used cephalosporins, whereas the latter frequently used macrolides and fluoroquinolones in addition to cephalosporins. In general, the causal bacteria of odontogenic infections can be treated with penicillins or penicillin/β-lactamase inhibitor combinations, with a single dose recommended before a procedure. Therefore, it may be beneficial for dentist-centered ASPs to focus on the appropriate use of oral cephalosporins in outpatients. The 2011 version of Japan's Guidelines for the Clinical Management of Infectious Diseases did not address odontogenic infections [27]. However, these were included in the 2014 version [28, 29] following the emergence of AMR as a major health threat. These guidelines recommend the use of penicillins, not cephalosporins, as first-line therapy for odontogenic infections [29]. Our analysis detected a slight increase in penicillin use accompanied by a minor decrease in cephalosporin use from 2015 to 2017, which may be indicative of a trend toward more appropriate AMU. A previous report suggested that dentists in Japan tend to follow the prescribing habits of more senior dentists without judicious consideration of current best practices [10]. However, we posit that the AMU among dentists will improve as the updated guidelines are increasingly adopted.

Our analysis showed that on-site dispensing accounted for 74.2% (DID: 0.89) of outpatient oral AMU among dentists, and that cephalosporins were the most frequently prescribed antimicrobials (66.9%). In Japan, approximately 80% of all dental facilities are privately owned and operated [30]. Our previous questionnaire-based study showed that the majority of private dental facilities are only able to provide on-site dispensing for two antimicrobials, one of which is frequently a cephalosporin [31]. We posit that the availability of two antimicrobials may be sufficient for most prophylactic or therapeutic applications in these facilities, even in cases where a patient is allergic to one of the antimicrobials. In consideration of these results, a potentially effective ASP for dentists may involve interventions to shift the use of cephalosporins to penicillins at private dental facilities. For example, antimicrobial prescribing has undergone a steady decline in England due to active interventions by the National Health Service, such as the release of antimicrobial stewardship messages that highlight the potential role of clindamycin in causing *Clostridioides difficile* infections [1]. Furthermore, the decline in antimicrobial prescribing is also expected to continue with the use of the Dental Antimicrobial Stewardship Toolkit [32]. This indicates that interventions aimed at dental facilities can improve antimicrobial prescription practices among dentists, and that standardized and intensive approaches may also be warranted in Japan.

This study has several limitations. First, the retrospective use of claims data means that we were unable to collect clinically relevant information regarding the prescriptions, such as patient and dentist characteristics, purpose of each prescription (therapeutic or prophylactic), and targeted diseases and procedures. Japan's Guidelines for the Clinical Management of Infectious Diseases [29] and the Japanese Circulation Society's 2017 Guidelines on the Prevention and Treatment of Infective Endocarditis [33] recommend the use of antimicrobials for infective endocarditis prophylaxis, which may represent one of the main reasons for

antimicrobial prescriptions by dentists. Nevertheless, we were unable to ascertain the specific reason for which each antimicrobial was prescribed. We are currently conducting a question-naire-based epidemiological study to further investigate the purpose of antimicrobial prescriptions by dentists. Second, the data were collected only from the NDB, which did not include uninsured patients such as tourists. However, these individuals would only account for a very small portion of the total population. Despite these limitations, our study's strength lies in the use of a national database that encompasses as much as 96% of the total population in Japan [12]. To the best of our knowledge, this is the first nationwide study of the epidemiology of AMU among Japanese dentists. In addition to dentists and physicians, AMU should also be examined in veterinarians as they act as both physicians and pharmacists in Japan and other countries. As the World Health Organization's One Health approach recommends a holistic and multisectoral approach to AMR measures, there is an urgent need to implement effective ASPs for both human and veterinary medicine [34].

In conclusion, this study showed that dentist-prescribed AMU mostly involved oral antimi-crobials for outpatients, and that cephalosporins were predominantly used. Dentists accounted for 9.1% of all outpatient oral AMU in Japan. There is a need for the development and imple-mentation of dentist-centered ASPs that focus on oral cephalosporins for outpatients. Further studies are also needed to identify the factors associated with dentists' prescription of oral cephalosporins in order to inform the development of more effective ASPs.

## Acknowledgments

We thank all the staff of the AMR Clinical Reference Center for their support in establishing Japan's National Action Plan on AMR from 2016 to 2020.

## Author Contributions

**Conceptualization:** Masahiro Ishikane, Yoshiki Kusama, Chika Tanaka, Sachiko Ono, Norio Ohmagari.

**Data curation:** Masahiro Ishikane, Yoshiki Kusama, Chika Tanaka.

**Formal analysis:** Masahiro Ishikane, Sachiko Ono, Shinya Tsuzuki.

**Funding acquisition:** Norio Ohmagari.

**Investigation:** Akane Ono, Masahiro Ishikane, Yoshiki Kusama, Chika Tanaka, Yuichi Mur-aki, Daisuke Yamasaki, Masaki Tanabe.

**Methodology:** Akane Ono, Masahiro Ishikane, Yoshiki Kusama, Chika Tanaka.

**Supervision:** Yuichi Muraki, Daisuke Yamasaki, Masaki Tanabe, Norio Ohmagari.

**Validation:** Yoshiki Kusama, Chika Tanaka.

**Writing – original draft:** Akane Ono, Masahiro Ishikane.

**Writing – review & editing:** Yoshiki Kusama, Chika Tanaka, Sachiko Ono, Shinya Tsuzuki, Yuichi Muraki, Daisuke Yamasaki, Masaki Tanabe, Norio Ohmagari.

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
