## [Decision Letter · Decision Letter 0]

10 Nov 2020

PONE-D-20-31740

The First National Survey of Antimicrobial Use Among Dentists in Japan from 2015 to 2017 Based on the National Database of Health Insurance Claims and Specific Health Checkups of Japan

PLOS ONE

Dear Dr. Kusama,

Thank you for submitting your manuscript to PLOS ONE. After careful consideration, we feel that it has merit but does not fully meet PLOS ONE’s publication criteria as it currently stands. Therefore, we invite you to submit a revised version of the manuscript that addresses the points raised during the review process.

Your manuscript was reviewed by two experts in the field, and they have requested some minor changes be made prior to acceptance.

If you could make these modifications and write a brief response to reviewers, that will greatly expedite review upon resubmission.

I wish you the best of luck with your revisions.

Hope you are keeping safe and well in these difficult times.

We look forward to receiving your revised manuscript.

Kind regards,

Simon Clegg, PhD

Academic Editor

PLOS ONE

"Yuichi Muraki received an honorarium for lecturing from Pfizer Japan, Inc. The other authors have no conflicts of interest to declare."

Reviewers' comments:

Reviewer's Responses to Questions

**Comments to the Author**

1. Is the manuscript technically sound, and do the data support the conclusions?

Reviewer #1: Yes

Reviewer #2: Yes

2. Has the statistical analysis been performed appropriately and rigorously? 

Reviewer #1: Yes

Reviewer #2: Yes

3. Have the authors made all data underlying the findings in their manuscript fully available?

Reviewer #1: Yes

Reviewer #2: Yes

4. Is the manuscript presented in an intelligible fashion and written in standard English?

Reviewer #1: Yes

Reviewer #2: Yes

5. Review Comments to the Author

Reviewer #1: This is an interesting study of the antibiotic prescribing practices of Japanese dentists. The study has made use of national data and the limitations of this are described. The findings are very interesting and while demonstrating a similar proportion of all antibiotic prescring by dentists in Japan to that in other countries, also shows some very big differences - particularly in the very high preference for prescribing cephalosporins.

This is such a striking difference, it perhaps warrants further highlighting. In most other countries where similar studies have been performed the most frequently prescribed antibiotics by dentists are penicillins (particularly amoxicillin). In the Americas this is often followed by clindamycin - particularly in those with penicillin allergy or in Europe, most often metronidazole - again particularly for those with penicillin allergy. In most countries where such studies of dental antibiotic use has been studied, cephalosporins come low in the list of those antibiotics prescribed. If possible, the reasons for the Japanese preference for cephalosporins should be discussed and there should be more discussion and comparison with other countries regarding the most widely used beta lactic alternatives. In that regard the authors should consider mention of a couple of relatively recent studies of dental antibiotic prescribing practice from Australia (Teoh L, Stewart K, Marino R J, McCullough M J. Part 1. Current prescribing trends of antibiotics by dentists in Australia from 2013 to 2016. Aust Dent J 2018) and the USA (Durkin M J, Hsueh K, Sallah Y H et al. An evaluation of dental antibiotic prescribing practices in the United States. J Am Dent Assoc 2017; 148: 878–886.)

It is also important to point out that antibiotic prescribing by dentists falls into two important categories (a) treatment of odontogenic infections) and (b) prophylaxis to prevent infection. And prophylaxis falls into two categories (i) prophylaxis to prevent local infection following dental procedures e.g. post-operative infection following a surgical extraction or implant placement and (ii) prophylaxis to prevent a distant site infection e.g. antibiotic prophylaxis (AP) to prevent infective endocarditis (IE) or prosthetic joint infection. One of the limitations of this type of study is that it cannot distinguish the reason/purpose for which the antibiotic was prescribed - and this should be mentioned in the limitations section. However, AP to prevent IE/prosthetic joint infections accounts for a significant % of all antibiotic prescribing by dentists in most countries and so this warrants mention. Particularly as the Japanese Cardiac Society issue recommendations to dentists that they provide AP to all patients at moderate or high-risk of IE undergoing invasive dental procedures (Nakatani S. et al, Circulation Journal, 2017). Most other countries only recommend AP for those at high-IE-risk, which is only around 10% of the number of individuals who are at moderate or high-risk. Therefore one would expect many more dental patients in Japan to be given AP to prevent IE than in other countries that recommend AP. Furthermore, the Japanese AP guidelines recommend the use of Amoxicillin 2g as a single oral dose 1 hour before the procedure or for those allergic to penicillin Clindamycin 600mg, Azithromycin 500mg or Clarithromycin 400mg. There is no recommendation for cephalosporins to be used orally for AP purposes. This raises questions about the compliance of Japanese dentists with the Japanese guidance on the use of AP to prevent IE and the need for interventions to address this.

I don't know if there is any recommendation for Japanese dentists to provide AP for patients with prosthetic joints but dentists in the US and some other countries are recommended to provide such AP.

The current categorisation of antibiotics into such broad categories seems rather a blunt instrument. Are the investigators not able to provide a breakdown of the types of penicillins - at least Amoxicillin v other penicillins to allow better comparison with the findings in other countries and a better understanding of compliance with Japanese AB and AP prescribing guidelines? Similarly, for comparison with other international studies data on the prescribing of clindamycin would be very helpful - if available

Reviewer #2: I had the fortune to review “The First National Survey of Antimicrobial Use Among Dentists in Japan from 2015 to 2017 Based on the National Database of Health Insurance Claims and Specific Health Checkups of Japan”.

My expertise is medical microbiology, dentistry and senior author of a comparable study in a different country.

With this background I have carefully read you very well performed, very well written, and very important study which should be published with priority.

A few recommendations to further improve this survey.

“Several countries have reported that dentists account for 7–11% of all AMU [1–6]. In the US, dentists constitute the second-leading prescribers of antimicrobials (10%) 60 after physicians (80%) [7],”

The message here is that physicians (by 80%) and dentists (by 10%) are THE prescribers of antibiotics and the address for need for stewardship. You might add a sentence about veterinarian and the fact that they are doctor and pharmacologist in personal union, at least in some countries. The stewardship should reach them also and quickly.

The second comment is related to this impressively high prescription rate of cephalosporins in Japan (above 60%). In other countries it is listed “among others” with only a few prescriptions at dental practices. To avoid such false prescriptions in the future, it is relevant to track the origin of problem. For instance, in our country clindamycin was (and is) a blockbuster in dentistry. The simple reason is the intense promotion by the corresponding manufacturer. That means, antibiotics of most attention (not of most efficacy) are prescribed. What could be the reason for such an over-prescription in Japan? It is known that there is cross-allergy between penicillins and cephalosporins (about 20%). My hypothesis is that the origin is related to allergy rates, namely that cephalosporins are generally regarded as safer? You might proof this hypothesis.

No other comments.

6. PLOS authors have the option to publish the peer review history of their article (what does this mean?). If published, this will include your full peer review and any attached files.

Reviewer #1: No

---

## [Author Response · Author response to Decision Letter 0]

7 Dec 2020

Reviewer #1’s Comments:

1. If possible, the reasons for the Japanese preference for cephalosporins should be discussed and there should be more discussion and comparison with other countries regarding the most widely used beta lactic alternatives.

Thank you for your thoughtful and constructive feedback regarding our manuscript.

We are grateful for this important suggestion. A possible reason for the preference for cephalosporins in Japan’s medical facilities may be the active promotion of these antimicrobials by pharmaceutical companies. A recent study reported that the top-selling antimicrobials in Nepal were those with a higher number of promotional activities [21]. Although such activities have since diminished in Japan, the situation was similar to that of Nepal until only 10 years ago. These promotions may have contributed to the high adoption rate of cephalosporins in medical facilities in Japan when compared to those in other countries. In fact, we have previously shown in an analysis of 31 Japanese hospitals that all the study hospitals had adopted the use of oral third-generation cephalosporins [22]. Furthermore, most of these cephalosporins were Japanese-origin drugs that are not commonly used outside of Japan. Based on the results of our previous questionnaire-based study on AMU among dentists in Japan [31], we are currently conducting a nationwide questionnaire-based epidemiological study to determine the factors that influence prescriptions of third-generation cephalosporins among 1700 dentists throughout the country. We have added this information to the Discussion section.

(Before: Discussion, Page 14, Line 201)

These results corroborate those of a previous cross-sectional questionnaire-based study on dentists at community clinics in Japan [20].

(After: Discussion)

These results corroborate those of a previous cross-sectional questionnaire-based study on dentists at community clinics in Japan [20]. The frequent use of cephalosporins in Japan may be due in part to the active promotion of these antimicrobials by pharmaceutical companies. A study conducted in Nepal showed that the top-selling antimicrobials were those that were subjected to intensive promotional activities [21]. Although such promotional activities have since diminished in Japan, the situation was similar to that of Nepal as recently as 10 years ago. These activities may therefore have contributed to the high adoption rate of cephalosporins in Japan’s medical facilities. In our previous analysis of AMU in 31 Japanese hospitals, we found that all the study hospitals had adopted the use of oral third-generation cephalosporins [22]. Furthermore, most of these cephalosporins were Japanese-origin drugs that are not commonly used outside of Japan [17].

References (References 21 and 22 have been added.)

17. Tsutsui A, Yahara K, Shibayama K. Trends and patterns of national antimicrobial consumption in Japan from 2004 to 2016. J Infect Chemother. 2018;24: 414-21. doi:10.1016/j.jiac.2018.01.003

20. Amari Y, Uehara Y, Watanabe Y, Inui A, Sugihara E, Yokokawa H, et al. Status of antimicrobial use among dentists in Japan. J Gen Hosp Med. 2014;6: 8-15. 

21. Koju P, Rousseau SP, Van der Putten M, Shrestha A, Shrestha R. Advertisement of antibiotics for upper respiratory infections and equity in access to treatment: a cross-sectional study in Nepal. J Pharm Policy Pract. 2020;13(4). doi:10.1186/s40545-020-0202-1

22. Kusama Y, Muraki Y, Mochizuki T, Kurai H, Gu Y, Ohmagari N. Relationship between drug formulary and frequently used cephalosporins, macrolides and quinolones in Japanese hospitals. J Infect Chemother. 2020;26(2):211-215. doi:10.1016/j.jiac.2019.08.013

31. Koizumi R, Kusama Y, Ishikane M, Tanaka C, Ono A, Gu Y, et al. Cross-sectional study to clarify the status of antimicrobial prescribing at outpatient care among dentists. Kansenshogaku Zasshi. (in press) 

2. In that regard the authors should consider mention of a couple of relatively recent studies of dental antibiotic prescribing practice from Australia (Teoh L, Stewart K, Marino R J, McCullough M J. Part 1. Current prescribing trends of antibiotics by dentists in Australia from 2013 to 2016. Aust Dent J 2018) and the USA (Durkin M J, Hsueh K, Sallah Y H et al. An evaluation of dental antibiotic prescribing practices in the United States. J Am Dent Assoc 2017; 148: 878–886.)

Thank you for your suggestion. We have added the suggested studies of dental antibiotic prescribing practice to the Discussion section.

(Before: Discussion, Page 14, Line 201) 

The prevalent use of cephalosporins by Japanese dentists is in contrast to the utilization patterns in Germany and the US [5, 21]. A national survey of German dentists showed amoxicillin to be the most commonly prescribed antimicrobial (45.8% of total AMU in 2015) [5], and a study of dental-related emergency department visits in the US revealed that penicillins accounted for over 60% of total AMU [21].

(After: Discussion)

The prevalent use of cephalosporins by Japanese dentists is in contrast to the utilization patterns in other countries [4, 5, 23-26]. For example, a national survey of German dentists showed amoxicillin to be the most commonly prescribed antimicrobial (45.8% of total AMU in 2015) [5]. A Canadian study using a population-based prescribing database reported on the dominant use of amoxicillin (1.26 DID out of 1.59 DID; 79.2% of total AMU in 2013) [4]. Similar findings have also been reported in Kuwait [23] and Scotland [24]. In comparisons of antimicrobial prescription rates, amoxicillin was also found to be the dominant antimicrobial in the US (57% in 2015) [25] and Australia (64.3% in 2016) [26].

References (References 23–26 have been added.)

4. Marra F, George D, Chong M, Sutherland S, Patrick DM. Antibiotic prescribing by dentists has increased Why? J Am Dent Assoc. 2016;147: 320-7. doi:10.1016/j.adaj.2015.12.014 

5. Halling F, Neff A, Heymann P, Ziebart T. Trends in antibiotic prescribing by dental practitioners in Germany. J Cranio-Maxillofacial Surg. 2017;45: 1854-9. doi:10.1016/j.jcms.2017.08.010 

23. Salako NO, Rotimi VO, Adib SM, Al-Mutawa S. Pattern of antibiotic prescription in the management of oral diseases among dentists in Kuwait. J Dent. 2004;32(7):503-509. doi:10.1016/j.jdent.2004.04.001

24. Roy KM, Bagg J. Antibiotic prescribing by general dental practitioners in the Greater Glasgow Health Board, Scotland. Br Dent J. 2000;188(12):674-676. doi:10.1038/sj.bdj.4800574

25. Durkin MJ, Hsueh K, Sallah YH, et al. An evaluation of dental antibiotic prescribing practices in the United States HHS Public Access. J Am Dent Assoc. 2017;148(12):878-886. doi:10.1016/j.adaj.2017.07.019

26. Teoh L, Stewart K, Marino RJ, McCullough MJ. Current prescribing trends of antibiotics by dentists in Australia from 2013 to 2016. Part 1. Aust Dent J. 2018;63(3):329-337. doi:10.1111/adj.12622

3. One of the limitations of this type of study is that it cannot distinguish the reason/purpose for which the antibiotic was prescribed - and this should be mentioned in the limitations section.

Thank you for the advice. We think that the major purpose of antimicrobials in our study setting is the prevention of infective endocarditis (IE). If oral medications are available, the Japanese Circulation Society Guidelines (2017) [33] recommend amoxicillin for IE prophylaxis; for patients with penicillin allergy, other antimicrobials such as clindamycin, azithromycin, or clarithromycin should be used. However, due to the nature of our database, we were unable to ascertain the detailed reason/purpose for which each antimicrobial was prescribed. To further investigate the reasons for prescribing antimicrobials by dentists, we are currently conducting a large-scale questionnaire-based epidemiological study. As advised, we have addressed this limitation in the revised manuscript.

(Before: Limitation, Page 16, Line 248) 

This study has several limitations. First, the retrospective use of claims data means that we were unable to collect clinically relevant information regarding the prescriptions, such as patient and dentist characteristics, purpose of each prescription (therapeutic or prophylactic), and targeted diseases and procedures. Second, the data were collected only from the NDB, which did not include uninsured patients such as tourists.

(After: Limitation) 

This study has several limitations. First, the retrospective use of claims data means that we were unable to collect clinically relevant information regarding the prescriptions, such as patient and dentist characteristics, purpose of each prescription (therapeutic or prophylactic), and targeted diseases and procedures. Japan’s Guidelines for the Clinical Management of Infectious Diseases [29] and the Japanese Circulation Society’s 2017 Guidelines on the Prevention and Treatment of Infective Endocarditis [33] recommend the use of antimicrobials for infective endocarditis prophylaxis, which may represent one of the main reasons for antimicrobial prescriptions by dentists. Nevertheless, we were unable to ascertain the specific reason for which each antimicrobial was prescribed. We are currently conducting a questionnaire-based epidemiological study to further investigate the purpose of antimicrobial prescriptions by dentists. Second, the data were collected only from the NDB, which did not include uninsured patients such as tourists.

References (Reference 33 has been added.)

29. Kaneko A, Aoki T, Ikeda F, Kawabe R, Satoh T, Tsumura N. The 2016 JAID/JSC guidelines for clinical management of infectious disease−Odontogenic infections. J Infect Chemother. 2018;24: 320-4. doi:10.1016/j.jiac.2017.09.014 

33. Nakatani S. Guideline on Prevention and Treatment of Infective Endocarditis. Circ J. 2019;83:1767-1809. doi:10.1253/circj.CJ-19-0549

4. The current categorisation of antibiotics into such broad categories seems rather a blunt instrument. Are the investigators not able to provide a breakdown of the types of penicillins - at least Amoxicillin v other penicillins to allow better comparison with the findings in other countries and a better understanding of compliance with Japanese AB and AP prescribing guidelines? Similarly, for comparison with other international studies data on the prescribing of clindamycin would be very helpful - if available.

Thank you for your suggestion. We agree that the current categorization is rather broad. If we collected more detailed data on the classification of antimicrobials, it may be possible to compare the use of amoxicillin and clindamycin (in addition to cephalosporins) with that of other countries. This may even help to identify the factors other than the promotional activities of pharmaceutical companies that explain why cephalosporins are the most commonly prescribed antimicrobial among Japanese dentists. However, our study was based on ATC Level 4 codes, which made it difficult to identify ampicillin and clindamycin use. The identification of ampicillin and clindamycin requires ATC Level 5 codes, in which amoxicillin is coded as J01CA04 and clindamycin is coded as J01FF01. The “penicillins” in our study comprised a combination of three sub-categories (J01CA, J01CE, and J01CR) at ATC Level 4; here, amoxicillin belongs to J01CA. The DID values of J01CA—including amoxicillin—were 0.10 in 2015, 0.10 in 2016, and 0.12 in 2017. These values were almost equivalent to those of “penicillins” in our study. Similarly, clindamycin belongs to J01FF at ATC Level 4. The DID values of J01FF—including clindamycin—were 0.0017 in 2015, 0.0019 in 2016, and 0.0022 in 2017. We have added these results to the Results section, and hope this helps provide insight for readers to make any comparisons if needed.

(Before: Result, Page 10, Line 155)

In contrast, the AMU trends for macrolides and fluoroquinolones were stable over time. Table 2 presents the DID values of total AMU among dentists stratified by administration route.

(After: Result)

In contrast, the AMU trends for macrolides and fluoroquinolones were stable over time. The penicillins in Table 1 included antimicrobials with ATC codes J01CA, J01CE, and J01CR. The DID values of J01CA antimicrobials (which include amoxicillin) were 0.10 in 2015, 0.10 in 2016, and 0.12 in 2017. The DID values of J01FF antimicrobials (which include clindamycin) were 0.0017 in 2015, 0.0019 in 2016, and 0.0022 in 2017. Table 2 presents the DID values of total AMU among dentists stratified by administration route.

Reviewer #2’s Comments:

1. You might add a sentence about veterinarian and the fact that they are doctor and pharmacologist in personal union, at least in some countries. The stewardship should reach them also and quickly. 

Thank you for your thoughtful and constructive feedback regarding our manuscript.

We agree with your suggestion. While physicians and dentists are the main ASP targets in human medicine, veterinarians are also a vital target for AMR in animal medicine under the WHO’s One Health approach. As advised, we have added the suggested information to the Discussion section with an additional reference.

(Before: Discussion, Page 17, Line 254)

To the best of our knowledge, this is the first nationwide study of the epidemiology of AMU among Japanese dentists. 

(After: Discussion)

To the best of our knowledge, this is the first nationwide study of the epidemiology of AMU among Japanese dentists. In addition to dentists and physicians, AMU should also be examined in veterinarians as they act as both physicians and pharmacists in Japan and other countries. As the World Health Organization’s One Health approach recommends a holistic and multisectoral approach to AMR measures, there is an urgent need to implement effective ASPs for both human and veterinary medicine [34].

References (Reference 34 has been added.)

34. World Health Organization. Global Action Plan on Antimicrobial Resistance. Vol 10.; 2015. Accessed November 25, 2020. https://www.who.int/antimicrobial-resistance/global-action-plan/en/

2. The second comment is related to this impressively high prescription rate of cephalosporins in Japan (above 60%). In other countries it is listed “among others” with only a few prescriptions at dental practices. To avoid such false prescriptions in the future, it is relevant to track the origin of problem. For instance, in our country clindamycin was (and is) a blockbuster in dentistry. The simple reason is the intense promotion by the corresponding manufacturer. That means, antibiotics of most attention (not of most efficacy) are prescribed. What could be the reason for such an over-prescription in Japan? It is known that there is cross-allergy between penicillins and cephalosporins (about 20%). My hypothesis is that the origin is related to allergy rates, namely that cephalosporins are generally regarded as safer? You might proof this hypothesis. 

Thank you very much for your comments and hypothesis. As you pointed out, cephalosporins may be prescribed if there are concerns for penicillin allergy. However, cephalosporins are not recommended as the primary alternative drug for such cases in Japan’s guidelines. We agree that a possible reason for the preference for cephalosporins in Japan’s medical facilities may be due to the active promotion of these antimicrobials by pharmaceutical companies. A recent study reported that the top-selling antimicrobials in Nepal were those with a higher number of promotional activities [21]. Although such activities have since diminished in Japan, the situation was similar to that of Nepal until only 10 years ago. These promotions may have contributed to the high adoption rate of cephalosporins in medical facilities in Japan when compared to those in other countries. In fact, we have previously shown in an analysis of 31 Japanese hospitals that all the study hospitals had adopted the use of oral third-generation cephalosporins [22]. Additionally, most of these cephalosporins were Japanese-origin drugs that are not commonly used outside of Japan. Nevertheless, we recognize that this is a hypothesis, and needs further empirical investigation. Therefore, based on the results of our previous questionnaire-based study on AMU among dentists in Japan [31], we are currently conducting a nationwide questionnaire-based epidemiological study to determine the factors (including the perceived safety of cephalosporins with respect to allergy rates) that influence prescriptions of third-generation cephalosporins among 1700 dentists throughout the country. We have added this information to the Discussion section.

(Before: Discussion, Page 14, Line 201)

These results corroborate those of a previous cross-sectional questionnaire-based study on dentists at community clinics in Japan [20].

(After: Discussion)

These results corroborate those of a previous cross-sectional questionnaire-based study on dentists at community clinics in Japan [20]. The frequent use of cephalosporins in Japan may be due in part to the active promotion of these antimicrobials by pharmaceutical companies. A study conducted in Nepal showed that the top-selling antimicrobials were those that were subjected to intensive promotional activities [21]. Although such promotional activities have since diminished in Japan, the situation was similar to that of Nepal as recently as 10 years ago. These activities may therefore have contributed to the high adoption rate of cephalosporins in Japan’s medical facilities. In our previous analysis of AMU in 31 Japanese hospitals, we found that all the study hospitals had adopted the use of oral third-generation cephalosporins [22]. Furthermore, most of these cephalosporins were Japanese-origin drugs that are not commonly used outside of Japan [17].

References (References 21 and 22 have been added.)

17. Tsutsui A, Yahara K, Shibayama K. Trends and patterns of national antimicrobial consumption in Japan from 2004 to 2016. J Infect Chemother. 2018;24(6):414-421. doi:10.1016/j.jiac.2018.01.003

20. Amari Y, Uehara Y, Watanabe Y, Inui A, Sugihara E, Yokokawa H, et al. Status of antimicrobial use among dentists in Japan. J Gen Hosp Med. 2014;6: 8-15. 

21. Koju P, Rousseau SP, Van der Putten M, Shrestha A, Shrestha R. Advertisement of antibiotics for upper respiratory infections and equity in access to treatment: a cross-sectional study in Nepal. J Pharm Policy Pract. 2020;13(4). doi:10.1186/s40545-020-0202-1

22. Kusama Y, Muraki Y, Mochizuki T, Kurai H, Gu Y, Ohmagari N. Relationship between drug formulary and frequently used cephalosporins, macrolides and quinolones in Japanese hospitals. J Infect Chemother. 2020;26(2):211-215. doi:10.1016/j.jiac.2019.08.013

31. Koizumi R, Kusama Y, Ishikane M, Tanaka C, Ono A, Gu Y, et al. Cross-sectional study to clarify the status of antimicrobial prescribing at outpatient care among dentists. Kansenshogaku Zasshi. (in press)

---

## [Editor Report · Decision Letter 1]

11 Dec 2020

The First National Survey of Antimicrobial Use Among Dentists in Japan from 2015 to 2017 Based on the National Database of Health Insurance Claims and Specific Health Checkups of Japan

PONE-D-20-31740R1

Dear Dr. Kusama,

We’re pleased to inform you that your manuscript has been judged scientifically suitable for publication and will be formally accepted for publication once it meets all outstanding technical requirements.

Kind regards,

Simon Clegg, PhD

Academic Editor

PLOS ONE

Additional Editor Comments:

Many thanks for resubmitting your manuscript to PLOS One

As all the comments have been addressed and the manuscript reads well, I have recommended it for publication

You should hear from the Editorial Office soon

It was a pleasure working with you, and I wish you all the best for your future research

Hope you are keeping safe and well in these difficult times

Thanks

Simon

---

## [Editor Report · Acceptance letter]

16 Dec 2020

PONE-D-20-31740R1 

The First National Survey of Antimicrobial Use Among Dentists in Japan from 2015 to 2017 Based on the National Database of Health Insurance Claims and Specific Health Checkups of Japan 

Dear Dr. Kusama:

I'm pleased to inform you that your manuscript has been deemed suitable for publication in PLOS ONE. Congratulations! Your manuscript is now with our production department. 

Kind regards, 

on behalf of

Dr. Simon Clegg 

Academic Editor

PLOS ONE